# Potential Role of PSMA-Targeted PET in Thyroid Malignant Disease: A Systematic Review

**DOI:** 10.3390/diagnostics13030564

**Published:** 2023-02-03

**Authors:** Alessio Rizzo, Manuela Racca, Sara Dall’Armellina, Roberto C. Delgado Bolton, Domenico Albano, Francesco Dondi, Francesco Bertagna, Salvatore Annunziata, Giorgio Treglia

**Affiliations:** 1Department of Nuclear Medicine, Candiolo Cancer Institute, FPO-IRCCS, 10060 Turin, Italy; 2Nuclear Medicine Unit, Department of Medical Sciences, AOU Città della Salute e della Scienza, University of Turin, 10126 Turin, Italy; 3Department of Diagnostic Imaging (Radiology) and Nuclear Medicine, University Hospital San Pedro and Centre for Biomedical Research of La Rioja, 26006 Logroño, Spain; 4Division of Nuclear Medicine, Università degli Studi di Brescia and ASST Spedali Civili di Brescia, 25123 Brescia, Italy; 5Unità di Medicina Nucleare, TracerGLab, Dipartimento di Diagnostica per Immagini, Radioterapia Oncologica ed Ematologia, Fondazione Policlinico Universitario A. Gemelli, IRCCS, 00168 Rome, Italy; 6Clinic of Nuclear Medicine, Imaging Institute of Southern Switzerland, Ente Ospedaliero Cantonale, 6501 Bellinzona, Switzerland; 7Faculty of Biology and Medicine, University of Lausanne, 1011 Lausanne, Switzerland; 8Faculty of Biomedical Sciences, Università della Svizzera Italiana, 6900 Lugano, Switzerland

**Keywords:** PET, nuclear medicine, PSMA, thyroid, thyroid cancer, imaging, systematic review

## Abstract

Background: Recently, several studies introduced the potential use of positron emission tomography/computed tomography (PET/CT) with prostate-specific membrane antigen (PSMA)-targeting radiopharmaceuticals in radioiodine-refractory thyroid cancer (TC). Methods: The authors accomplished a comprehensive literature search of original articles concerning the performance of PSMA-targeted PET/CT in TC patients. Original papers exploring this molecular imaging examination in radioiodine-refractory TC patients undergoing restaging of their disease were included. Results: A total of 6 documents concerning the diagnostic performance of PSMA-targeted PET/CT in TC (49 patients) were included in this systematic review. The included articles reported heterogeneous values of PSMA-targeted PET/CT detection rates in TC, ranging from 25% to 100% and overall inferior to [^18^F]-fluorodeoxyglucose PET/CT when the two molecular imaging examinations were compared. Two studies reported the administration of [^177^Lu]PSMA-radioligands with theragnostic purpose in three patients. Conclusions: The available literature data in this setting are limited and heterogeneous. The employment of PET with PSMA-targeting radiopharmaceuticals in this setting did not affect patient management. Nevertheless, prospective multicentric studies are needed to properly assess its potential role in TC patients.

## 1. Introduction

Thyroid cancer (TC) is the most common endocrine malignancy, and its incidence keeps growing yearly worldwide. This increase has been attributed to improved screening practices, including neck ultrasound and fine-needle aspiration biopsy of small thyroid nodules [1]. Most newly diagnosed TCs are represented by small and asymptomatic papillary TCs, belonging to a sizeable subclinical reservoir of indolent tumors, which would likely have remained unknown for all patients’ lifetime in most cases [2]. Nevertheless, the increase of newly diagnosed TCs, often termed “overdiagnosis,” concerns small papillary TCs, as well as high-risk TCs, voluminous tumors with advanced stage at diagnosis, gross extra-thyroidal extension, and aggressive histopathological subtypes [3]. In this context, the recently observed growth in mortality rates among patients with advanced-stage TC suggests aggressive post-surgical treatments and accurate risk stratification. Post-therapeutic whole-body scan after radioiodine (RI) administration has historically played an essential role in evaluating the tumor load and RI avidity of residual or recurrent disease, as well [3]. Unfortunately, only about two-thirds of TC metastatic patients demonstrate RI uptake in their lesions, whereas the remaining develop metastases that do not show significant RI uptake (or lose the capability to concentrate it) or have progressive disease after RI treatment [3,4]. This finding induced authors to introduce the concept of RI-refractoriness, which is defined, according to the latest American and European Thyroid and Nuclear Medicine Societies, as: no RI concentration on a diagnostic or post-therapeutic RI scan in patients with abnormal thyroglobulin (Tg) levels or evidence of disease in other instrumental diagnostic examinations; the presence of [^131^I]I uptake in some, but not in all, tumor foci; progressive disease, despite evidence of [^131^I]I uptake in TC lesions; and TC metastases progression, despite a cumulative administered [^131^I]I activity above 22.2 GBq [5].

In the clinical setting of RI-refractory TC, cross-sectional imaging with contrast-enhanced computed tomography (CT) of the neck, chest, and abdomen is the first-line instrumental examination to assess the presence of locally recurrent invasive disease (and its relationship with vessels), lymph node metastases in neck and mediastinum regions, as well as distant metastases in lung and bones [3]. Furthermore, CT is the main instrument to assess the response to treatment with tyrosine-kinase inhibitors (TKI) through the “Response Evaluation Criteria in Solid Tumors” (RECIST) [6].

Concerning molecular imaging examinations other than RI scintigraphy, Fluorine-18 fluorodeoxyglucose ([^18^F]FDG) positron emission tomography (PET)/CT is the most studied hybrid imaging method in RI-refractory TC patients. According to guidelines, its execution is appropriate in TC patients with elevated serum Tg and a negative post-therapeutic whole-body RI scan [3]. Nevertheless, its accuracy is affected by several features, including tumor dedifferentiation and burden, and it has a superior detection rate (DR) in patients with aggressive histological subtypes. Moreover, high [^18^F]F-FDG uptake on PET images is an independent prognostic factor for overall survival in TC patients [7,8].

Prostate-specific membrane antigen (PSMA), a transmembrane protein encoded by the gene FOLH1 [9,10], is a relatively novel target for molecular imaging and therapy in prostate cancer imaging [11,12]. Nevertheless, recent literature reported that this receptor is often expressed on the cell membrane of neovascular endothelial cells of different solid tumors other than prostate cancer [13]. This should be a basis for employing PSMA-targeted PET as a diagnostic tool in tumors other than prostate cancer, including TC.

Several recent studies evaluated the performance of PET imaging with PSMA-targeting radiopharmaceuticals in TC. This paper aims to accomplish a systematic review concerning PSMA-targeted PET/CT performance in patients with this group of malignancies. As a secondary purpose, this paper aims to collect evidence comparing diagnostic performance between PET with PSMA-radioligands and other imaging examinations in TC.

## 2. Materials and Methods

### 2.1. Protocol

The present systematic review was conducted following a predefined protocol [14], and the “Preferred Reporting Items for a Systematic Review and Meta-Analysis” (PRISMA 2020 statement) were used as a basis for its writing [15]. The complete PRISMA checklist is available as Appendix A.

Firstly, a review question was defined: can PET/CT with PSMA-targeting radiopharmaceuticals detect TC lesions?

The Population, Intervention, Comparator, Outcomes (PICO) framework was employed as a basis for the literature search. The criteria for study eligibility were established as follows: patients with diagnosed TC (Population) submitted to PSMA-targeted PET (Intervention) compared with molecular or conventional imaging (Comparator). The accuracy of PSMA-targeted PET/CT in TC patients and the PSMA-radioligand uptake in TC lesions were defined as outcomes.

Three reviewers (A.R., S.D.A., and G.T.) independently performed the literature search, the study selection, the quality assessment, and the data extraction. An online consensus meeting solved any discrepancies among the reviewers.

### 2.2. Literature Search Strategy and Information Sources

The above-mentioned authors, as already stated, performed a comprehensive literature search using two electronic bibliographic databases (Cochrane library and PubMed/MEDLINE), seeking papers that explored PSMA-targeted PET imaging diagnostic performance in TC patients.

The search algorithm was based on a combination of these terms: (A) “PET” OR “positron” AND (B) “PSMA” AND (C) “thyroid.” No restrictions were applied concerning the article’s language or publication year. Moreover, reviewers screened included studies’ references searching for additional eligible articles. Finally, the ClinicalTrials.gov database was consulted to report ongoing studies.

The literature search was last updated on 25 December 2022.

### 2.3. Eligibility Criteria

Clinical studies reporting data about PSMA-targeted PET imaging in TC patients were considered suitable for inclusion in this systematic review. Editorials, letters, reviews, comments, case reports, or small case series on the topic were excluded from qualitative analysis, just as original studies concerning different issues (including pre-clinical studies).

### 2.4. Selection Process

The above-mentioned authors independently screened the titles and abstracts of the papers and selected the studies eligible for the systematic review based on the predefined inclusion and exclusion criteria, specifying the reason for all the decisions.

### 2.5. Data Collection Process and Data Extraction

Three authors extracted data from all the included studies in the full text, tables, and figures concerning general study information (authors, publication year, country, study design, and funding sources); patient characteristics (sample size, age, sex ratio, clinical setting, histological TC subtypes, and thyroglobulin levels); index text characteristics (PSMA-radioligand employed, type of hybrid imaging protocol, administered activity, uptake time between radiopharmaceutical administration and image acquisition, and the protocol for image analysis), and comparative diagnostic imaging.

### 2.6. Quality Assessment (Risk of Bias Assessment)

The selected method used for assessing the risk of bias in individual studies and the applicability to the review question was QUADAS-2, a tool for evaluating quality in diagnostic test accuracy studies [16]. Three reviewers assessed the studies’ grades in the systematic review in four domains (patient selection, index test, reference standard, and flow and timing) concerning the risk of bias and in three fields regarding the applicability (patient selection, index test, and reference standard).

## 3. Results

### 3.1. Literature Search and Study Selection

The literature search was updated to 25 December 2022 and provided 76 records. On the basis of the inclusion and exclusion criteria stated in the Materials and Methods section, 70 papers were excluded (24 as not in the field of interest; 18 as reviews, editorials, book chapters, or letters; and 28 as case reports). Consequently, six articles were eligible for inclusion in the systematic review (qualitative synthesis) after the full-text assessment [17,18,19,20,21,22]. Reviewers could not find additional suitable papers screening the references of these articles. Figure 1 summarizes the study selection process.

### 3.2. Study Characteristics

The analysis of the characteristics concerning the 6 papers included in the systematic review (qualitative research), including 49 TC patients, are presented in Table 1, Table 2 and Table 3 [17,18,19,20,21,22]. Regarding general study data (Table 1), the included papers were published between 2017 and 2022 in Europe, the USA, India, and Mexico. Half of the studies were conducted with a prospective design [17,19,20], whereas the remaining half was retrospective. All the studies included in this systematic review were monocentric [17,18,19,20,21,22].

Regarding the patient key characteristics (Table 2), the number of patients ranged from 5 to 11 (median age from 49 to 72 years; male percentage from 20% to 75%) [17,18,19,20,21,22]. All the included studies employed PSMA-targeted PET/CT to analyze patients with RI-refractory TC [17,18,19,20,21,22]. When reported, median thyroglobulin levels ranged from 22 to 2482 ng/mL [17,19,20,21].

Among the included studies, the performance of PSMA-targeted PET/CT was explored in different histopathological subtypes; the most representative TC subtypes were papillary and follicular TC, with a total of 28 and 9 patients enrolled, respectively [17,18,19,20,21]. In all papers but one, PSMA-targeted PET/CT performance was compared to [^18^F]F-FDG PET/CT [17,18,19,20,22], whereas in the remaining study, the comparative imaging was [^131^I]I scintigraphy [21]. Concerning the patients’ preparation before the [^18^F]F-FDG PET/CT scan, none of the included studies reported whether the exam was performed under TSH stimulation [17,18,19,20,22].

Several heterogeneities were observed among the papers included in this systematic review concerning the index test key characteristics (Table 3). Because of this, a meta-analysis could not be performed. In all the included articles, the PSMA-targeting radiopharmaceutical employed was [^68^Ga]Ga-PSMA-11 (activity range: 83–212 MBq in absolute values) [17,19,20,21,22]. All the included studies but one coregistered PET images with low-dose CT [17,18,20,21,22], whereas in the remaining one, PET images were coregistered with magnetic resonance imaging (MRI) [19]. The uptake time between PSMA-radioligands injection and PET acquisition ranged from 60 to 73 min. Qualitative and semiquantitative analyses were performed in all the articles included in this systematic review. The main semiquantitative values calculated were the mean and maximal standardized uptake values (SUV_max_ and SUV_mean_, respectively) [17,18,19,20,21,22]; only one paper added the target-to-background uptake ratio (TBR) assessment by dividing lesions’ SUV_max_ for the pectoralis major muscle’s SUV_max_ [21].

### 3.3. Risk of Bias and Applicability

Taking advantage of the data reported in each study, the authors assessed the risk of bias and concerns about the applicability of the included papers based on the QUADAS-2 instruments. The results of the quality assessment are reported in Figure 2.

### 3.4. Results of Individual Studies (Qualitative Synthesis)

The assessment of the accuracy of PET/CT with PSMA-targeting radiopharmaceuticals showed relatively weak diagnostic performance in 4 of the studies included in the systematic review, with a detection rate (DR) ranging from 25% to 83% on a per-patient-based analysis and about 65% on a per-lesion-based analysis [17,19,20,22]. Only 2 studies reported a detection rate of 100% on the per-patient-based analysis [18,21]; nevertheless, one of them used an [^131^I]I scan as comparative imaging, which has low sensitivity in the setting of RI refractory TC [21].

No adverse effects were reported in the included studies after the injection of PSMA-targeting radiopharmaceuticals for diagnostic purposes.

Concerning the semiquantitative evaluation of TC lesions, significant heterogeneity was found in SUV_max_ reported values of the PET-positive lesions in the included studies; indeed, it ranged from 1.0 to 39.7 for metastatic lesions. Five of the included papers reported that PET with PSMA-targeting radiopharmaceuticals had an overall lower diagnostic performance than [^18^F]F-FDG PET, despite, in some cases, it could show lesions without significant glucose metabolism [17,18,19,20,22].

With regard to the accuracy of PET imaging with PSMA-targeting radiopharmaceuticals in different TC histopathological subtypes, none of the included studies made a statistical analysis to assess differences in uptake values among the explored histopathological variants. Overall, slightly higher uptake was reported in follicular TC, whereas low or absent uptake was described in dedifferentiated TC lesions [18,19,22]. When reported, the uptake values of TC lesions on PET/CT images did not correlate to the grade of PSMA staining at immunohistochemistry analysis, particularly in dedifferentiated TC, where PSMA staining was observed on the immunohistochemistry analysis without significant uptake on PET/CT images [22].

Since the only diagnostic PSMA-targeting radiopharmaceutical employed in all the studies was [^68^Ga]Ga-PSMA-11, there are no reports available concerning the differences in the accuracy of the different PSMA radiopharmaceuticals available in this clinical setting.

Finally, two studies reported the employment of [^177^Lu]Lu-PSMA-617 in three RI-refractory TC patients with theragnostic purpose [18,22]. Two of the included patients had a slight and temporary response to the treatment, followed by an increase in serum Tg levels and progressive disease after a few months, whereas the remaining showed disease progression one month after the treatment. Concerning the side effects encountered after the administration of the therapy, temporary nausea (grade not reported) was observed after the second cycle in one patient [22].

The results of the included papers, including semiquantitative metrics, sites of the lesions, and DRs, are reported in Table 4.

## 4. Discussion

The overexpression of PSMA on the cell membrane of prostate cancer cells make it a suitable target for molecular imaging and radioligand therapy in prostate cancer patients [23]. This discovery allowed the development of various radiolabeled, PSMA-binding, low-weight molecules to increase these patients’ diagnostic and therapeutic options. Subsequent studies observed that PSMA is also expressed in tumors other than prostate cancer, including TC, clear cell renal cancer, and hepatocellular carcinoma [24]. However, conversely from what was observed in the prostate cancer lesions microenvironment, these neoplasms show PSMA expression in tumor-associated neovascular endothelium instead of cancer cells. Moreover, immunohistochemical studies reported that an intense PSMA staining in the neovasculature of several TC subtypes, including papillary TC and follicular TC, was correlated to more clinically aggressive behavior than that with weak PSMA expression; in particular, they observed that patients whose lesions had moderate to strong PSMA expression were more likely to develop RI-refractoriness or die of TC [25,26,27]. Interestingly, studies focused on the immunohistochemical analyses reported that anaplastic TC, despite its well-known aggressiveness, showed a lower PSMA expression than well-differentiated TC [25]; this result may be explained by the underlying biological progression pathway expressed by this TC subtype, which usually shows lower microvessel density than well-differentiated TC [28].

To corroborate the hypothesis that the employment of PET with PSMA-targeting radiopharmaceuticals might play a role in TC patients’ management, several case reports and retrospective studies reported a low incidence of thyroid incidentalomas in patients undergoing PSMA-targeted PET/CT for other purposes (generally prostate cancer restaging) [29,30]. These reports assessed that focal PSMA-radioligands uptake could be seen both in benign and malignant thyroid nodules since both conditions are associated with the presence of neoangiogenesis [31]; however, since only one study reported a higher uptake of PSMA-radioligands in malignant lesions than in benign ones [32], more studies are needed to assess the potential of PSMA-targeted PET/CT in discriminating malignant thyroid nodules.

The presence of both PSMA staining in TC lesion samples and PSMA-radioligands uptake in thyroid incidentalomas was a worthy rationale, inspiring several authors to explore a potential role for PSMA-targeted PET imaging in patients with TC and to compare its performance to the actual standard of care. As the main topic concerning TC is the management of RI-refractory disease, all the reviewed studies enrolled patients in this clinical setting [17,18,19,20,21,22].

Since most of the papers included in this systematic review were pilot studies investigating PSMA-targeted PET imaging performance in a limited number of TC patients (49 in six studies), the emerged evidence seems limited and quite heterogeneous. Of particular interest is the variability observed in the DR reported in each study, ranging from 25 to 100% [17,18,19,20,21,22]; moreover, a significant heterogeneity among the values of SUV_max_ was observed in the per-lesion analysis of every single paper. These findings may be primarily explained by the low number of included patients, the different comparator imaging employed, and the heterogeneity concerning the histological TC subtypes analyzed. Neoangiogenesis in TC is modulated by distinct signaling pathways, and no study correlated the presence (or absence) of PSMA-targeting radiopharmaceuticals uptake to the histology or the mutational status of TC lesions. Further, it is key to harmonize PET/CT procedures in order to make results comparable between different centers and at different time-points [33]. In this setting, more studies are warranted to understand the biological mechanisms underlying the discrepancies reported.

[^18^F]F-FDG PET/CT is the recommended hybrid imaging method of choice in patients with RI [3]. When compared to [^18^F]F-FDG PET/CT, PSMA-targeted PET imaging could detect a lower number of lesions, and it was not able to change patient management [17,18,19,20,22]. For this reason, based on the available literature, it cannot be currently suggested as a valid alternative imaging technique to restage patients with RI-refractory TC, taking into account evidence-based data.

One of the included studies tried to correlate the presence of uptake on PSMA-targeted PET/CT images to the expression of PSMA by the neovascular endothelium cells on histological samples in seven out of the eight enrolled patients [22]. The authors found concordance between in vitro and in vivo examinations in three patients and discordance in the remaining four. Based on the available literature data, this finding, previously described also in other malignancies expressing PSMA in their neovasculature (including clear cell renal cancer, hepatocellular carcinoma, and glioblastoma [34,35,36]), still cannot be explained; in this context, further studies are needed to assess the biological mechanisms underlying this phenomenon.

To date, RI-refractory TC can be treated with molecular-driven antiangiogenic therapies consisting of TKIs, including lenvatinib [37,38], which is currently approved by the Food And Drug Administration (FDA) for the treatment of RI-refractory TC, since it is able to prolong progression-free survival in this kind of patient by inhibiting the pathways through vascular-endothelium growth factor receptor [39]. Most patients undergoing this treatment demonstrated disease stabilization or partial response within 12–24 months after the initiation of the therapy [40]. Moreover, this antiangiogenic pharmaceutical does not require tumor mutational profiling since it can also be prescribed when no specific targetable mutations have been discovered. In this context, PSMA-targeted PET/CT used as a biomarker of neoangiogenesis might play a role in predicting which lesions are more likely to respond to TKI treatment, despite its lower sensitivity while compared to the current standard of care. Since none of the included studies tried to assess the prognostic value of PSMA-targeting radiopharmaceuticals uptake in patients undergoing systemic therapy with TKI, prospective trials are warranted to explore its potential role in this setting.

Finally, the most exciting potential employment of PSMA-targeting radiopharmaceuticals relies on theragnostics. Based on the outstanding results reported by the VISION trial concerning the administration of [^177^Lu]Lu-PSMA-617 in patients with metastatic castration-resistant prostate cancer [11], it is feasible that tumors characterized by PSMA expression (even if in the neovasculature instead of on the cancer cells) might present a satisfactory response to this therapy. In this context, PSMA-targeted PET/CT should have a role in assessing which patients are suitable to undergo radioligand therapy since only patients with significant uptake in all the known lesions could be eligible. Among the studies included in this systematic review, two assessed the efficacy of [^177^Lu]Lu-PSMA-617 treatment in three RI-refractory TC patients [18,22]. Two patients showed a transient response to the radioligand therapy, followed by biochemical progression after a few months, whereas the remaining showed disease progression one month after the first cycle of treatment. Overall, data on the theragnostic potential of PSMA-targeting radiopharmaceuticals in TC are currently insufficient to justify using PSMA-targeted PET in TC. Further clinical and pre-clinical investigations are needed to clearly assess the potential role of PSMA-targeted theragnostics in TC patients.

To the best of our knowledge, this is the first systematic review concerning the employment of PSMA-targeting radiopharmaceuticals in patients with TC. However, this manuscript has several limitations: most of the included papers were pilot studies with poor sample sizes and, consequently, a significant patient selection bias. Furthermore, the included articles show considerable heterogeneity in the results. For these reasons, a meta-analysis was not feasible.

## 5. Conclusions

Even if some studies suggested the potential use of PSMA-targeted PET/CT in TC, the available data in this setting are limited and heterogeneous.

Overall, a significant advantage in terms of the diagnostic accuracy of PSMA-targeted PET/CT compared to [^18^F]F-FDG PET/CT in RI-refractory TC was not demonstrated.

The potential advantage of the theragnostic value of PSMA-targeted radiopharmaceuticals needs to be further demonstrated (the RI-refractory TC patients performing PSMA radioligand therapy in the literature are scarce).

As suggestions for future studies, prospective multicentric studies comparing PSMA-targeted PET/CT with [^18^F]F-FDG PET/CT in RI-refractory TC and studies comparing PSMA-targeted PET findings with immunohistochemical data are warranted.

## Figures and Tables

**Figure 1 diagnostics-13-00564-f001:**
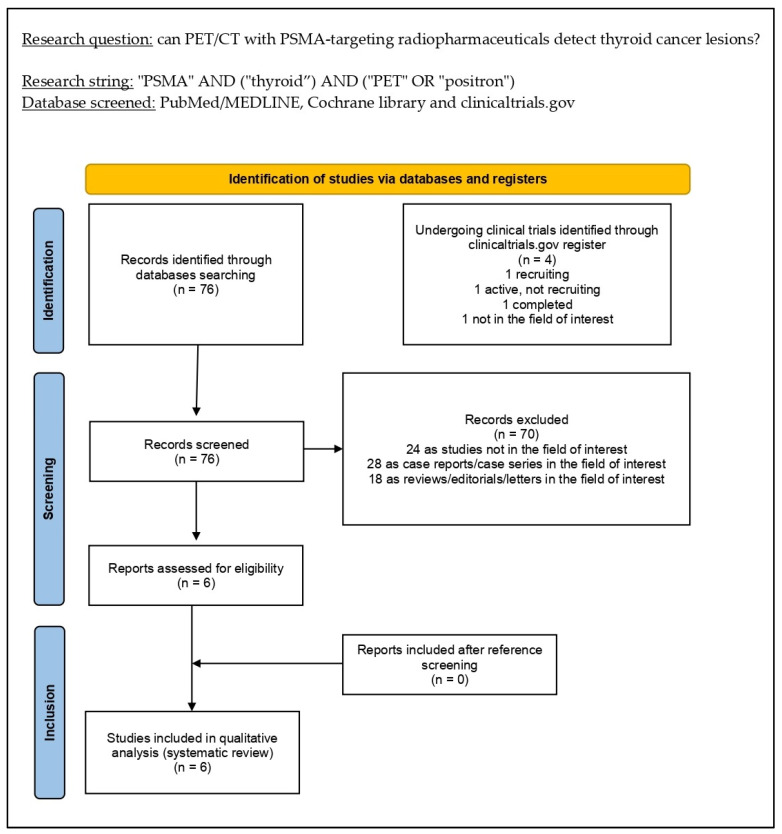
Comprehensive overview of the study selection process for the systematic review.

**Figure 2 diagnostics-13-00564-f002:**
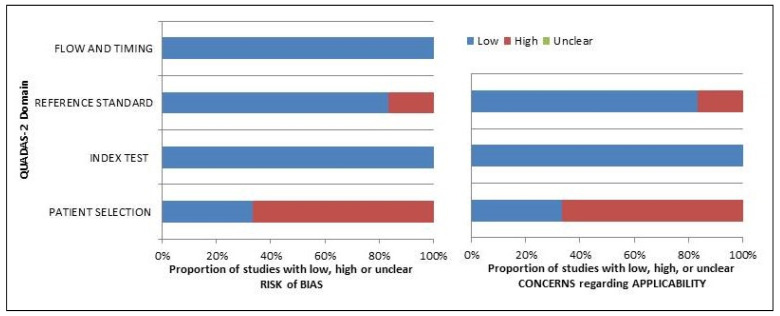
Quality assessment according to QUADAS-2 tool. Authors classified the papers included in the systematic review as high risk or low risk of bias or applicability concerns for distinct domains listed in the ordinate axis. The abscissa axis shows the percentage of studies. The graph indicates that over 60% of the included studies are affected by a high risk of bias.

**Table 1 diagnostics-13-00564-t001:** General study information.

Authors [Ref.]	Year	Country	Study Design/Number of Involved Centers	Funding Sources
Lütje et al. [17]	2017	Germany	Prospective/Monocentric	None declared
De Vries et al. [18]	2020	The Netherlands	Retrospective/Monocentric	None declared
Lawhn-Heath et al. [19]	2020	USA	Prospective/Monocentric	Several grants declared
Verma et al. [20]	2021	India	Prospective/Monocentric	None declared
Pitalua-Cortes et al. [21]	2021	Mexico	Retrospective/Monocentric	None declared
Wächter et al. [22]	2022	Germany	Retrospective/Monocentric	Several grants declared

**Table 2 diagnostics-13-00564-t002:** Patient key characteristics and clinical settings.

Authors [Ref.]	Sample Size	Median Age (Years)	Gender (Male %)	Clinical Setting (No. Patients)	Histopathological TC Subtypes (No. Patients)	Median Thyroglobulin (ng/mL)	Comparative Imaging
Lütje et al. [17]	6	72	n.a.	RI-refractory TC	2 papillary 4 follicular	2482	[^18^F]F-FDG PET/CT
De Vries et al. [18]	5	50	20%	RI-refractory TC	5 papillary	n.a.	[^18^F]F-FDG PET/CT (4 patients)
Lawhn-Heath et al. [19]	11	65	45%	RI-refractory TC	5 papillary 2 follicular 2 Hürtle cell 2 anaplastic	22	[^18^F]F-FDG PET/CT; [^123^I]I/[^131^I]I scan
Verma et al. [20]	9	49	66.6%	RI-refractory TC	9 papillary	225	[^18^F]F-FDG PET/CT
Pitalua-Cortes et al. [21]	10	58	20%	RI-refractory TC	7 papillary 3 follicular	773	[^131^I]I scan
Wächter et al. [22]	8	59	75%	Anaplastic or dedifferentiated TC	6 poorly differentiated 2 anaplastic	n.a.	[^18^F]F-FDG PET/CT

Legend: CT: computed tomography; FDG: fluorodeoxyglucose; n.a.: not available; PET: positron emission tomography; RI: radioiodine; TC: thyroid cancer.

**Table 3 diagnostics-13-00564-t003:** Index test key characteristics.

Authors [Ref.]	Tracer	Hybrid Imaging	Tomograph	Administered Activity	Uptake Time (Minutes)	Image Analysis
Lütje et al. [17]	[^68^Ga]Ga-PSMA-11	PET/CT	Biograph mCT (Siemens^®^)	Range: 91–93 MBq	62 ± 7	Qualitative and semiquantitative (SUV_max_ and SUV_mean_)
De Vries et al. [18]	[^68^Ga]Ga-PSMA-11 [^177^Lu]Lu-PSMA-617	PET/CT	TruePoint Biograph mCT40 (Siemens^®^)	Range: 1.5–2 MBq/kg	60	Qualitative and semiquantitative (SUV_max_)
Lawhn-Heath et al. [19]	[^68^Ga]Ga-PSMA-11	PET/MRI	Time-of-fight Signa (GE ^®^)	Median: 207.2 MBq	73	Qualitative and semiquantitative (SUV_max_)
Verma et al. [20]	[^68^Ga]Ga-PSMA-11	PET/CT	Unspecified time-of-flight tomograph (Philips ^®^)	Mean: 83 MBq	60	Qualitative and semiquantitative (SUV_max_)
Pitalua-Cortes et al. [21]	[^68^Ga]Ga-PSMA-11	PET/CT	Biograph mCT20 (Siemens^®^)	Range: 148–185 MBq	60	Qualitative and semiquantitative (SUV_max_ and TBR)
Wächter et al. [22]	[^68^Ga]Ga-PSMA (not further specified) [^177^Lu]Lu-PSMA-617	PET/CT	n.a.	Range: 140–212 MBq	60	Qualitative and semiquantitative (SUV_max_)

Legend: CT: computed tomography; MRI: magnetic resonance imaging; PET: positron emission tomography; PSMA: prostate-specific membrane antigen; SUV: standardized uptake value; TBR: target-to-background ratio; MBq = MegaBecquerel.

**Table 4 diagnostics-13-00564-t004:** Outcomes of the included studies.

Authors [Ref.]	Lesions SUV_max_	Lesions Site	Detection Rate
Lütje et al. [17]	Range: 3.3–39.7	local recurrence lymph node bone soft tissue	Per patient: 83.3%
De Vries et al. [18]	Range: 0.85–10.56	lymph nodes lungs liver leptomeningeal	Per patient: 100%
Lawhn-Heath et al. [19]	Range: 1.0–27.8	local recurrence lymph node lungs bone	Per patient: 72.7% Per lesion: 65.1%
Verma et al. [20]	Range: 10.1–45.67	Lungs bone	Per patient: 55% Per lesion: 64%
Pitalua-Cortes et al. [21]	Range: 1.8–70.5	lymph nodes lung bone brain	Per patient: 100%
Wächter et al. [22]	Range: 1.3–6.3	local recurrence lymph node lung bone	Per patient: 25%

Legend: SUV_max_: maximal standardized uptake value.

## Data Availability

The data presented in this study are available on request from the corresponding author.

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
