# Peer review of "Potential Role of PSMA-Targeted PET in Thyroid Malignant Disease: A Systematic Review"

_diagnostics, 2023, doi:10.3390/diagnostics13030564_

Round 1

Reviewer 1 Report

Dear Authors.

The article is interesting and I enjoyed reading it. Thyroid cancer represents a major topic in endocrinology, nuclear medicine, and endocrine surgery, as well from a public health perspective due to its epidemiological impact. That is why new diagnostic options in severe and challenging cases might bright useful clues in thyroid malignancy – associated overall approach.

Here are my observations which I consider as minor suggestions:

1.    Abstract – I suggest a few hard data (number) based on your systemic review should be provided in Abstract in addition to the number of papers and number of patients you identified

2.    Introduction – Line 70 – Please cite a reference concerning “RECIST”

3.    Results – Did the included patients in those studies have a histological confirmation with regarding to the mentioned imaging scans?

4.    Discussion – Since there are only a few studies regarding this particular matter with a small number of patients, do you think that incorporating in the final analyze, for instance, at Discussion the case reports you already identify would bring some value to our current understanding, for instance, the data from Table 2:

Verburg, F.A. 2015

Damle, N.A 2016

Taywade, S.K 2016

Derlin, T  2017

Jena, A.; 2017

Kirchner, J. 2017

Damle, N.A.; 2018

Sasikumar, A. 2018

Singh, D.; 2018

Ciappuccini, R 2019

Gossili, F.; 2020

Santhanam, P 2020

Sood, A.; 2020

Tang, K.; 2020

Tupalli, A 2020

Ciappuccini, R.; 2021

Civan, C 2021

Li, H.; 2021

Sonavane, S.N.; 2021

Usmani, S.; 2021

Verma, P 2021

Freesmeyer, M.; 2022

Hasenauer, N 2022

Lu, Y. 2022

Parghane, R.V.; 2022

Rosar, F.; 2022

Thank you,

Best regards,

Author Response

Thanks to the reviewer for the comment and the suggestions.

  1. Abstract – I suggest a few hard data (number) based on your systemic review should be provided in Abstract in addition to the number of papers and number of patients you identified

         According to reviewer suggestions, more data concerning the reported values of detection rate were provided in the Abstract section.

  1. Introduction – Line 70 – Please cite a reference concerning “RECIST”

         We added the reference concerning RECIST as requested.

  1. Results – Did the included patients in those studies have a histological confirmation with regarding to the mentioned imaging scans?

         Most of the included studies did not histologically confirm the malignant nature of the reported lesions, and 5/6 compared the results of PSMA-targeted PET/CT with [18F]FDG PET/CT imaging or, in case of mismatch in doubtful lesions, with MRI. The only study that compared PSMA-targeted PET data with histology/immunohistochemistry was Wächter et al. and their results are reported both in results and discussion sections.

  1. Discussion – Since there are only a few studies regarding this particular matter with a small number of patients, do you think that incorporating in the final analyze, for instance, at Discussion the case reports you already identify would bring some value to our current understanding, for instance, the data from Table 2:

         As stated by the reviewer, only a few studies explored the potential role of PSMA-targeted PET/CT in managing RI-refractory thyroid cancer. Nevertheless, case reports were excluded from the analysis (as stated in the material and method section) since they are a potential source of publication bias (indeed, even including only original studies, we highlighted a significant patient selection bias in the results section due to the low number of included patients).

Reviewer 2 Report

This is a well-executed, well-written comprehensive systematic review on the value of the PSMA ligand in differentiated thyroid cancer. It is an interesting topic although not very original (the authors mention multiple other reviews). It summarises the finding of 6 original papers.

Major:
- if SUVmax’es are mentioned, the authors should state whether this was measured using EARL-compliant protocols.

Minor:
- no white line before paragraph 2.2 and 2.3
- 2 times pararagraph 2.2 and 2 times paragraph 3, check numbering
- no white line between table 2 and “several heterogeneities …”
- typo in table 2: Hurtle >> Hürthle
- comparative imaging >> reference imaging?
- “a slight incidence” >> “a low incidence”?
- discussion: consider discussing the role of TSH stimulation in FDG-PET and mentioning if it was done for reference imaging

Author Response

Thanks to the reviewer for the comment and the suggestions.

- if SUVmax’es are mentioned, the authors should state whether this was measured using EARL-compliant protocols

No data about EARL are reported in the included studies.

- no white line before paragraph 2.2 and 2.3
- 2 times pararagraph 2.2 and 2 times paragraph 3, check numbering
- no white line between table 2 and “several heterogeneities …”
- typo in table 2: Hurtle >> Hürthle
- comparative imaging >> reference imaging?
- “a slight incidence” >> “a low incidence”?
- discussion: consider discussing the role of TSH stimulation in FDG-PET and mentioning if it was done for reference imaging

We rephrased as requested by the reviewer.

Concerning the patients' preparation before the [18F]F-FDG PET/CT scan, none of the included studies reported whether the exam was performed under TSH stimulation. We added this sentence in the results section.
